# Colorectal Pulmonary Metastases: Pulmonary Metastasectomy or Stereotactic Radiotherapy?

**DOI:** 10.3390/cancers15215186

**Published:** 2023-10-28

**Authors:** Martijn van Dorp, Constantia Trimbos, Wilhelmina H. Schreurs, Chris Dickhoff, David J. Heineman, Bart Torensma, Geert Kazemier, Frank J. C. van den Broek, Ben J. Slotman, Max Dahele

**Affiliations:** 1Amsterdam University Medical Center, Location VUmc, Department of Cardiothoracic Surgery, 1007 MB Amsterdam, The Netherlands; 2Cancer Center Amsterdam, Imaging and Biomarkers, 1081 HV Amsterdam, The Netherlands; 3Northwest Clinics, Department of Surgery, 1815 JD Alkmaar, The Netherlands; 4Department of Anesthesiology, Leiden University Medical Centre, 2333 ZA Leiden, The Netherlands; 5Amsterdam University Medical Center, Location VUmc, Department of Surgery, 1007 MB Amsterdam, The Netherlands; 6Máxima Medical Centra, Department of Surgery, 5657 EG Veldhoven, The Netherlands; 7Amsterdam University Medical Center, Location VUmc, Department of Radiation Oncology, 1007 MB Amsterdam, The Netherlands

**Keywords:** colorectal lung metastases, oligometastatic colorectal cancer, pulmonary metastasectomy, SABR, RCT

## Abstract

**Simple Summary:**

Patients with oligometastatic colorectal pulmonary metastases have similar overall survival after radical-intent stereotactic ablative radiotherapy or pulmonary metastasectomy. The local recurrence rate for patients after stereotactic radiotherapy with oligometastatic colorectal pulmonary metastases was 38% with long-term follow-up and 52% of these patients were able to undergo local salvage therapy. Even though all patients underwent local treatment for oligometastatic disease, 46% of patients from both groups would eventually develop polymetastatic conversion. The randomized controlled COPPER trial is currently being designed in the Netherlands and will randomize patients with lung-limited oligometastatic colorectal pulmonary metastases between stereotactic radiotherapy and metastasectomy.

**Abstract:**

Background: Pulmonary metastasectomy and stereotactic ablative radiotherapy (SABR) are both guideline-recommended treatments for selected patients with oligometastatic colorectal pulmonary metastases. However, there is limited evidence comparing these local treatment modalities in similar patient groups. Methods: We retrospectively reviewed records of consecutive patients treated for colorectal pulmonary metastases with surgical metastasectomy or SABR from 2012 to 2019 at two Dutch referral hospitals that had different approaches toward the local treatment of colorectal pulmonary metastases, one preferring surgery, the other preferring SABR. Two comparable patient groups were identified based on tumor and treatment characteristics. Results: The metastasectomy group comprised 40 patients treated for 69 metastases, and the SABR group had 60 patients who were treated for 90 metastases. Median follow-up was 38 months (IQR: 26–67) in the surgery group and 46 months (IQR: 30–79) in the SABR group. Median OS was 58 months (CI: 20–94) in the metastasectomy group and 70 months (CI: 29–111) in the SABR group (*p* = 0.23). Five-year local recurrence-free survival (LRFS) was 44% after metastasectomy and 30% after SABR (*p* = 0.16). Median progression-free survival (PFS) was 15 months (CI: 3–26) in the metastasectomy group and 10 months (CI: 6–13) in the SABR group (*p* = 0.049). Local recurrence rate was 12.5/7.2% of patients/metastases respectively after metastasectomy and 38.3/31.1% after SABR (*p* < 0.001). Lower BED Gy_10_ was correlated with an increased likelihood of recurrence (*p* = 0.025). Clavien Dindo grade III-V complication rates were 2.5% after metastasectomy and 0% after SABR (*p* = 0.22). Conclusion: In this retrospective cohort study, pulmonary metastasectomy and SABR had comparable overall survival, local recurrence-free survival, and complication rates, despite patients in the SABR group having a significantly lower progression-free survival and local control rate. These data would support a randomized controlled trial comparing surgery and SABR in operable patients with radically resectable colorectal pulmonary metastases.

## 1. Background

Pulmonary metastasectomy and stereotactic ablative radiotherapy (SABR; also known as stereotactic body radiotherapy, SBRT) are both guideline-supported treatments for radical-intent local therapy of pulmonary oligo-metastases in selected colorectal cancer (CRC) patients. However, there is no randomized controlled study comparing local therapies, and robust criteria for selecting one treatment modality over the other in fit, medically operable patients, with radically resectable disease, are lacking. While this is reflected in the UK NICE guidelines, which concluded that there was insufficient evidence to recommend one local therapy over another for colorectal pulmonary metastases (CRPM) amenable to local treatment [1], other guidelines consider surgery as the standard of care. The European Society for Medical Oncology (ESMO) guideline favors surgery if a lesion is considered completely resectable and suggests considering SABR if surgery would be widely invasive, as an addition to surgery to achieve a complete treatment, or as an alternative if patients are inoperable due to frailty or poor anatomical location [2]. The North American expert consensus document on pulmonary metastasectomy favors surgery and suggests that SABR should be considered in patients who are not suitable for surgery, who decline resection, or who develop ipsilateral metastases after prior metastasectomy, although they prefer thermal ablation over SABR for smaller lesions [3]. The Chinese expert consensus document also favors surgery, suggesting that radiotherapy should be considered in patients not fit for, or declining surgery, with a tumor location unsuitable for surgery, or inadequate predicted postoperative lung function [4].

Stereotactic radiotherapy offers an ambulatory, noninvasive option with a favorable quality-of-life profile without the short-term risks of surgery [5]. SABR may also spare functional lung parenchyma in patients who are at risk of future metastases. Given the lack of high-quality evidence in favor of local therapy for CRPM [6], many large-volume centers treat both operable and non-operable patients with SABR because of the limited risk for complications. Several retrospective comparative studies are available comparing surgical metastasectomy and SABR. These studies mainly compare patients with unresectable disease (for various reasons) to those with resectable disease, and yet they describe comparable overall survival rates [7,8,9,10], even though a clear difference in local control rate can be observed in favor of surgical metastasectomy [11,12].

The aim of this study was to compare surgery and SABR in patients with CRPM eligible for local treatment. We compared outcomes from two Dutch referral hospitals with different approaches to the local treatment of CRPM: one preferred pulmonary metastasectomy, and the other SABR. Patients were selected to allow for a homogenous cohort regarding tumor and treatment characteristics.

## 2. Methods

### 2.1. Patient Selection

Amsterdam University Medical Center (AUMC) and Máxima Medical Center (MMC) had different treatment approaches for patients with oligometastatic CRPM: AUMC favored SABR, and MMC favored surgery. Patients that received local treatment for CRPM from January 2012 to December 2019 in these two centers were selected for review (Medical Ethics Review Committee of the AUMC and MMC, 2021.0733). Patients without written consent for retrospective chart review were contacted for approval. Chart review was performed independently by two authors (MvD and CT). Castor Electronic Data Capture (EDC) was used for data management (https://www.castoredc.com (accessed on 18 February 2022). During the study period, 65 patients underwent pulmonary metastasectomy at MMC, and 144 patients underwent SABR for CRPM at AUMC. Several criteria were used to select two homogenous cohorts in both hospitals (Figure 1). Patients were excluded if they were treated for oligoprogression after previous palliative chemotherapy for polymetastatic conversion (n = 29), did not have prior (radical) treatment of the primary tumor (n = 4), were previously treated for other oligometastases except colorectal liver metastases (CRLM) (n = 10), had previously received other local treatment for CRPM (n = 32), were simultaneously treated with SABR and metastasectomy for multiple pulmonary metastases (n = 5), did not have CT follow-up available (e.g., no access to follow-up imaging elsewhere) (n = 24). In the metastasectomy group, patients were also excluded if, after pathological analysis, the lesion was not a CRC metastasis (n = 6; e.g., non-small cell lung cancer, granuloma, focal atelectasis). During the period analyzed, six patients underwent metastasectomy as initial treatment of CRPM at the AUMC, and no patients underwent SABR for CRPM at MMC.

### 2.2. Outcome

The primary endpoint was overall survival (OS) and was defined as the time from local therapy to death from any cause or last follow-up. Local recurrence-free survival (LRFS) was defined as the time from local therapy to local recurrence, or death from any cause or last follow-up. Progression-free survival (PFS) was defined as the time from local therapy to the first radiological progression or death from any cause or last follow-up. Local recurrence following SABR was defined as the development of a new metastasis in the planning target volume in the same lobe, which corresponded to hypermetabolic areas on fluorodeoxyglucose-positron emission tomography (FDG-PET) CT and/or was histologically confirmed. Local recurrence following metastasectomy was defined as a new metastasis adjacent to the staple line, which corresponded to hypermetabolic areas on fluorodeoxyglucose-positron emission tomography (FDG-PET) CT and/or was histologically confirmed. Pulmonary recurrence was defined as new pulmonary metastasis, not defined as local recurrence. Polymetastatic conversion (PMC) was defined as the development of new metastases (in any organ) not amenable to further local-only treatment based on multidisciplinary tumor board (MTB) recommendation.

For purposes of comparing different radiation dose-fractionation prescriptions, they were expressed as the biologically effective dose (BED Gy_10_, i.e., using an α/β ratio of 10 for tumor effect). Complications following local therapy were classified using the Clavien-Dindo classification [13]. The survival period was measured from the date of surgery or start of SABR to the date of death from any cause or the last date of data extraction (18 February 2022).

### 2.3. Statistical Analysis

Continuous normally distributed variables are presented as means and standard deviation (SD) or 95% confidence intervals (CI), continuous non-normally distributed variables by their median and interquartile range (IQR). Categorical variables are presented as frequencies with percentages. Continuous normally distributed variables were tested with the Student’s *t*-test for independent samples. Non-normally distributed data were tested with the Mann-Whitney U-test for independent samples. Categorical variables were tested using the Pearson’s χ^2^ test or Fisher’s exact test as appropriate. Missing data were determined to be at random before precluding data from analysis. The Pearson correlation coefficient was used to examine the relationship between different normally distributed variables, and the Spearman rank test was used for non-normally distributed variables. Kaplan-Meier survival analysis was performed to determine the difference in survival. Statistical significance was set at *p* ≤ 0.05. All analyses were conducted using SPSS statistics (Version 28.0).

## 3. Results

### 3.1. Patient Selection

A total of 100 patients met the inclusion criteria; 40 patients were treated in MMC and underwent pulmonary metastasectomy, and 60 patients were treated in AUMC and received SABR. Patient characteristics and treatment characteristics of the primary tumor were comparable between the two groups (Table 1 and Appendix A). Mean age of patients from both groups was 67 years, and 90% of all patients had an Eastern Cooperative Oncology Group (ECOG) performance score of less than two. Seventeen patients (42.5%) in the metastasectomy group (median 3 CRLM) and 28 patients (46.7%) in the SABR group (median 2 CRLM), were previously treated for colorectal liver metastases.

### 3.2. Local Therapy

In the metastasectomy group, 69 metastases were treated, with a median of 1 per patient (range 1–5). Sixty percent of patients were treated for a solitary metastasis. The median size of the metastases on imaging was 12 mm (IQR: 9–19), and the median size after pathological analysis was 12 mm (IQR: 8–19). One patient underwent a bilobectomy, 11 metastases were removed by means of a lobectomy, 56 wedge resections (81% of procedures) were performed, and 1 segmentectomy was performed. A minimally invasive resection was performed in 36 patients (90%), and no mediastinal lymph node dissection was performed. Ten patients (25%) were treated for bilateral metastases; eight patients with a staged resection, and two patients by means of a simultaneous resection. A complete (R0) resection was achieved in all patients. One patient had an intrapulmonary lymph node metastasis.

In the SABR group, 90 metastases were treated, with a median of 1 per patient (range 1–4). Sixty-three percent of patients were treated for a solitary metastasis. The median size of the metastases on imaging was 12 mm (IQR: 9–20). SABR dose fractionation was as follows: 1 × 34 Gy (n = 1 metastasis), 3 × 18 Gy (n = 19), 5 × 11 Gy (n = 36), 8 × 7.5 Gy (n = 27), 12 × 5 Gy (n = 6), 30 × 2 Gy (n = 1). The median BED Gy_10_ was 116 (IQR 105–151). No patient underwent SABR for thoracic lymph node involvement. Thirteen patients (21.0%) were treated for bilateral metastases.

The 90-day Clavien-Dindo grade III–V complication rate was 2.5% (n = 1) in the metastasectomy group and 0% in the SABR group (*p* = 0.22). This patient underwent a bronchoscopy for postoperative atelectasis. Ninety-day mortality was 0% in the metastasectomy group and 1.6% (n = 1) in the SABR group (*p* = 0.42). This patient with ECOG performance status of 3 underwent SABR for a solitary CRPM and developed a urinary tract infection soon after SABR, but the exact cause of death was unknown.

### 3.3. Outcome

In the metastasectomy group, the median follow-up was 38 months (IQR 26–67) and 46 months (IQR 30–79) in the SABR group (*p* = 0.195). Median OS was 58 months (CI: 20–94) in the metastasectomy group and 70 months (CI: 29–111) in the SABR group (*p* = 0.23). The estimated median LRFS was 58 months (CI: 20–95) in the metastasectomy group and 29 months (CI: 23–36) in the SABR group (*p* = 0.16). The 5-year LRFS was 44% in the metastasectomy group and 30% in the SABR group (*p* = 0.16) (Figure 2). Median PFS was 15 months (CI: 3–26) in the metastasectomy group and 10 months (CI: 6–13) in the SABR group (*p* = 0.049). Five-year PFS was 30% in the metastasectomy group and 14% in the SABR group (Figure 3). The local recurrence rate was 12.5% of patients in the metastasectomy group, and 38.3% in the SABR group (*p* < 0.001) (Table 2).

### 3.4. Treatment of Local Failure

In the metastasectomy group, five patients (12.5%) and 5/69 metastases (7.2%) developed a local recurrence (all after R0 wedge resection; there was no significant difference in size between metastases that failed and did not fail after resection). Median time to local recurrence was 13 months (IQR: 6–27) (Figure 4). Two other patients developed an isolated regional lymph node recurrence, and one patient developed a port-site chest wall recurrence. All five patients underwent salvage treatment for local failure, and this included one repeat resections (local control rate of 0%), SABR in three patients (local control rate 0%), and one patient underwent percutaneous radiofrequency ablation (local control rate of 0%). Two patients received palliative treatment for local recurrence.

In the SABR group, 23 patients (38.3%) and 28/90 metastases (31.1%) developed a local recurrence. Median time to local recurrence was 19 months (IQR: 12–30). Two other patients developed an isolated regional lymph node recurrence. The median size of the metastases that failed after SABR was significantly larger than those that did not (16 mm [IQR 9–24] vs. 11 mm [IQR 7–15], *p* = 0.037). Higher fractionation and lower BED Gy_10_ were correlated with an increased likelihood of recurrence (r = −0.24, *p* = 0.025). BED > 125 Gy_10_ demonstrated a numerically lower risk of local failure (19% vs. 41%, *p* = 0.08). Thirteen patients (52%) underwent salvage treatment for local failure; this included a salvage resection in eight patients (local control rate of 100%), reirradiation in three patients (local control rate of 0%), and percutaneous ablation in two patients (local control rate of 50%). Seventeen patients received palliative treatment for local recurrence.

In both groups combined, a total of 19 patients underwent a salvage treatment (resection, SABR, or ablation) for local failure with a median follow-up of 33 months (IQR 21–71) following local salvage. The survival rate was significantly lower if the salvage treatment again resulted in local failure compared to the patients in which the salvage treatment successfully obtained local control (30% vs. 100%, *p* = 0.006) (Figure 5).

### 3.5. Treatment of New Metastases

In the metastasectomy group, 17 patients (42.5%) developed new CRPM. Four patients (23.5%) underwent a total of 11 more local treatments, and thirteen patients received palliative therapy for PMC. In the SABR group, 28 patients (46.7%) developed new pulmonary metastases during follow-up. Fifteen patients (53.6%) underwent a total of 24 local treatments, and thirteen patients received palliative therapy for PMC. Significantly more patients underwent another local treatment for new pulmonary metastases in the SABR group compared to the surgery group (46.7% vs. 23.5%, respectively, *p* = 0.048).

In the metastasectomy group, 50.0% of patients developed PMC, and 37.5% never developed another metastasis after the initial metastasectomy. In the SABR group, 43.3% of patients developed PMC, and 20% never developed another metastasis. Overall survival was significantly shorter for patients with PMC (26 months (CI: 18–33) compared to patients without PMC (93 months (CI: 78–109) (*p* < 0.001).

## 4. Discussion

This report demonstrates long-term outcomes in carefully selected patients from two expert centers where patients were treated with either surgical metastasectomy or SABR for colorectal pulmonary metastases. Our main findings were that the primary outcome, overall survival, was not significantly different between the metastasectomy group and the SABR group. Regarding secondary outcome measures, LRFS and the rate of postprocedural complications were also not significantly different between both groups. However, 5-year PFS was significantly lower for patients treated with SABR (14%) compared to metastasectomy (30%), and the local recurrence rate was significantly higher after SABR (31% of metastases) compared to metastasectomy (7% of metastases). We acknowledge the wide confidence intervals in the data, suggesting that a larger sample may be needed for more definitive outcome analysis. Both groups had an even distribution of prognostic factors. Therefore, no statistical matching was performed in this analysis. In the metastasectomy group, 13% of patients were excluded because the final pathology did not reveal colorectal metastasis. Such patients may have been included in the SABR group, as only 25% of patients in the SABR group had pathologically proven colorectal pulmonary metastases.

Clinical data indicate that the number of patients with oligometastatic disease receiving aggressive treatment is increasing rapidly [14,15,16,17]. Many observational studies have shown good long-term survival with minimal morbidity after radical-intent local therapy for pulmonary metastases [16,18,19], and the randomized SABR-COMET trial demonstrated significant improvement in overall survival in patients treated with SABR for a variety of primary tumor types and metastases [20]. Although the control group of the PulMiCC trial revealed that patients with limited colorectal pulmonary metastases have better survival than previously assumed without metastasectomy, with a 5-year overall survival of 22% [21], the recently published ESTS survey on pulmonary metastasectomy noted that 92% of surgeons believe that pulmonary metastasectomy results in improved survival for colorectal cancer patients [22]. Challenges in interpreting the data from this current study include a substantially lower local control rate for SABR, a similar PMC rate for both groups, and no OS difference, despite the driving factor for treating oligometastatic disease with local therapy being that local tumor control will improve survival [23,24]. The favorable OS in the SABR group, despite the high local failure rate, might be influenced by the aggressive use of radical-intent local salvage therapy in more than half of the patients with local failure. Successful local salvage demonstrated a significantly improved overall survival compared to patients with persistent local failure. This highlights the importance of close follow-up strategies and of discussing patients with possible local failure in a thoracic oncology tumor board attended by surgeons experienced with salvage procedures. The relatively low local control rate after SABR for colorectal lung metastases reported here is consistent with prior data and is also consistent with reports in other organs such as the liver [11,25]. This report also confirms the hypothesis that a high BED is associated with a lower failure rate for colorectal pulmonary metastases [26], and found that more fractionated SABR schedules resulted in higher local failure rates. This suggests that when a lesion cannot be treated with a high BED, surgery should be given serious consideration.

To date, four other original reports have been published comparing metastasectomy to SABR for pulmonary metastases with variable patient selection and outcome. An analysis from South Korea of 51 patients with metastases from various primary tumors and a median follow-up of 13 months demonstrated no difference in two-year OS, PFS, and local control rates between both groups [7]. A Dutch propensity-matched report of 110 patients with metastases from various primary tumors reported comparable 5-year OS, PFS, and local control rates between both groups [8]. An Italian study of 142 metastasectomy and 28 SABR patients treated for colorectal pulmonary metastases reported similar two-year OS between both groups [9]. A propensity-matched analysis from MD Anderson in 381 patients with colorectal pulmonary metastases and a median follow-up of 53 months found a significantly higher risk of local recurrence after SABR: 29% vs. 14% and 37% vs. 18% for SABR vs. wedge resection within 2 and 5 years respectively; OS was not presented [10]. The results from these reports highlight the importance of long-term follow-up in two homogenous cohorts, especially regarding the distribution of known prognostic factors and tumor histology. The data suggest that local control following SABR is different depending on the histology of the pulmonary metastases [27,28]. This should be considered in this analysis. Furthermore, sufficient follow-up is required to demonstrate differences in local control rates between different treatment modalities; an apparent difference in local control rate at five years might not be detectable after only two years of follow-up. This report is distinguished from other reports by including only patients with colorectal cancer pulmonary metastases; by the long-term follow-up, detailed outcome parameters (including salvage interventions), and by using matched patient cohorts from two hospitals, thereby reducing the risk of selection bias regarding the local treatment, and resulting in two homogenous cohorts without significant differences between prognostic factors.

In addition to factors already mentioned, several other limitations are important to note. An analysis of 100 patients might not suffice to demonstrate differences in OS rates between the groups. Improvements in systemic therapy over time may influence parameters like OS in future cohorts. Moreover, because many patients were referred by other centers, details on the use of (subsequent) systemic treatments were lacking, which could have impacted survival rates. Other reports have also demonstrated the difficulty of comparing the effect of both local and systemic treatments for colorectal pulmonary metastases [29]. These two highly selected cohorts had no significant difference in known prognostic factors but were treated by different MTBs. We acknowledge the possibility of selection bias with this study design; also, the definition of PMC might have been used differently, and a significant difference was seen in the rate of repeat local interventions for recurrent pulmonary metastases. All of these factors could have affected the rate of PMC and survival rates. Lastly, in most patients, no (K)RAS, BRAF, or microsatellite instability biomarkers were determined. With respect to the low level of histological proof of colorectal cancer origin in the SABR group, all lesions had been deemed to be consistent with colorectal metastases following MTB review, and we note that all salvage surgery cases confirmed colorectal histology.

## 5. Conclusions

In conclusion and as highlighted by various guidelines [1,2,3,4], there is a lack of high-quality comparative studies between different local treatment modalities for patients with colorectal pulmonary metastases. In this retrospective cohort study, pulmonary metastasectomy and SABR resulted in comparable OS, LRFS, and complication rates. Patients in the SABR group had a significantly lower PFS and local control rate. These data support a randomized controlled trial comparing surgery and SABR. We are presently designing a multicenter prospective randomized controlled trial in which patients with colorectal pulmonary oligometastases will be randomized to minimally invasive metastasectomy or SABR for three or less colorectal pulmonary metastases (NCT05808790) [30]. In the absence of randomized data, and with a need to make practical recommendations based on the available data, the local recurrence rate after SABR suggests that when a sufficiently high BED cannot be delivered using SABR or when post-SABR salvage surgery is considered difficult, surgery should be given serious consideration.

## Figures and Tables

**Figure 1 cancers-15-05186-f001:**
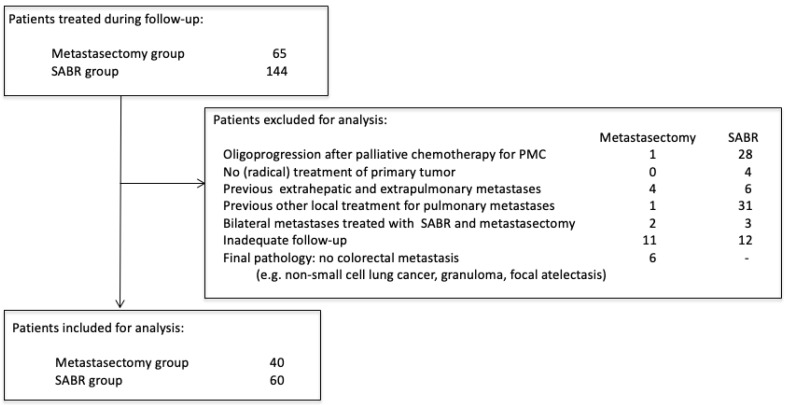
Patient selection criteria.

**Figure 2 cancers-15-05186-f002:**
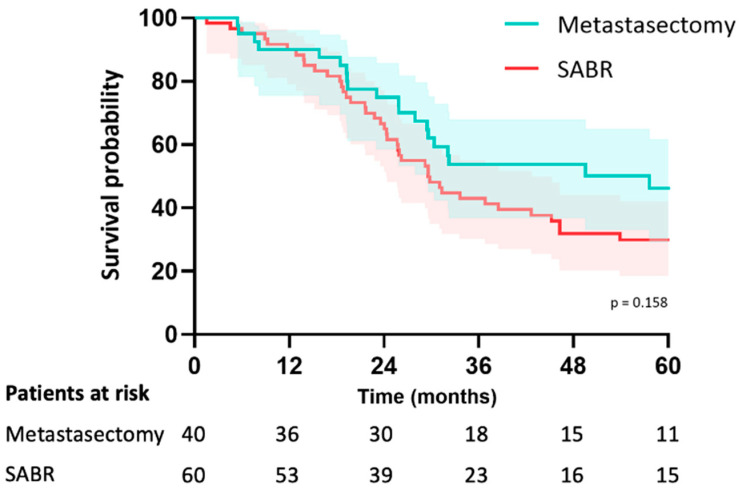
Kaplan-Meier curve for local recurrence-free survival of patients with colorectal pulmonary metastases after metastasectomy or SABR.

**Figure 3 cancers-15-05186-f003:**
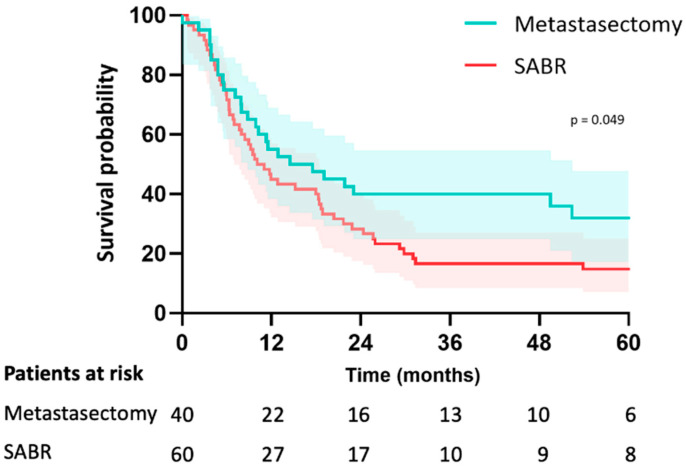
Kaplan-Meier curve for PFS of patients with colorectal pulmonary metastases after metastasectomy or SABR.

**Figure 4 cancers-15-05186-f004:**
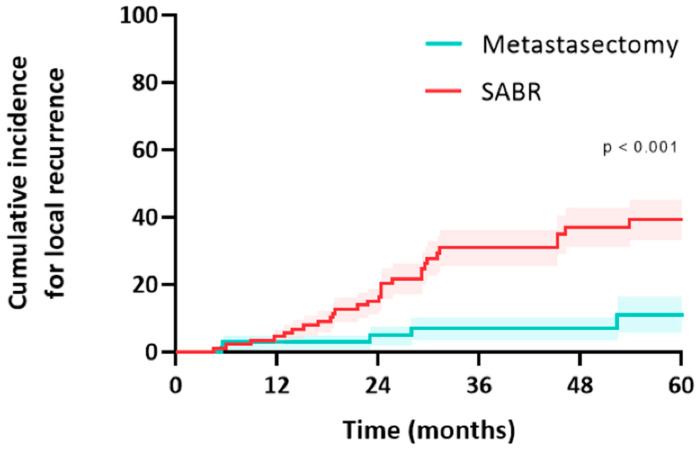
Cumulative incidence for local recurrence of colorectal pulmonary metastases after metastasectomy or SABR.

**Figure 5 cancers-15-05186-f005:**
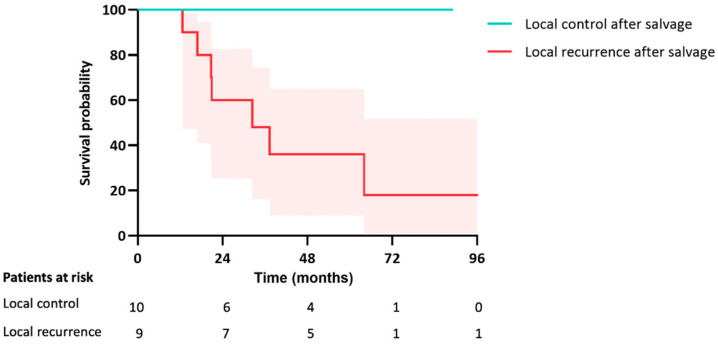
Kaplan-Meier curve for OS after salvage treatment for local failure in both groups.

**Table 1 cancers-15-05186-t001:** Patient and tumor characteristics.

	Metastasectomy (n = 40)	SABR (n = 60)	*p* Value
Age (yr)	67 ± 9	67 ± 10	0.48
Sex (% male)	25 (62.5%)	40 (66.7%)	0.59
ECOG 0–1 (%)	37 (92.5%)	52 (86.7%)	0.38
Charlson comorbidity index	1 (0–2)	0 (0–1)	0.06
Primary right-sided colon cancer	6 (15.0%)	13 (21.7%)	0.29
Primary left-sided colon cancer	15 (37.5%)	14 (23.3%)	
Primary rectal cancer	19 (47.5%)	33 (55.0%)	
Synchronous lung metastases	3 (7.5%)	6 (10.0%)	0.71
Metachronous lung metastases	37 (92.5%)	54 (90.0%)	
Disease free interval (months)	21 (12–39)	20 (10–36)	0.71
Number of lung metastases	1 (1–2)	1 (1–2)	0.10
Total number of lung metastases	69	90	
Patients with solitary pulmonary metastases	24 (60%)	38 (63%)	0.83
Size of lung metastases (mm)	12 (9–19)	12 (9–20)	0.33
Bilateral metastases	10 (25%)	12 (20%)	0.63
Lung-limited pulmonary metastases	23 (57.5%)	32 (53.3%)	0.79
Pre-treatment pathologically proven CRPM	5 (12.5%)	15 (25.0%)	0.13
Patients with previously treated CRLM	17 (42.5%)	28 (46.7%)	
Number of previously treated CRLM	3 (1.5–5.5)	2 (1–5)	0.62

Data are median (IQR), mean ± SD, or n (%). ECOG = Eastern Cooperative Oncology Group. CRPM = colorectal pulmonary metastases. CRLM = colorectal liver metastases.

**Table 2 cancers-15-05186-t002:** Secondary outcome measures.

	Metastasectomy	SABR	*p* Value
Median LRFS	58 months (CI: 20–94)	70 months (CI: 29–111)	0.16
Median PFS	15 months (CI: 3–26)	10 months (CI: 6–13)	0.049
Complication rate	2.5%	0%	0.22
LRR per metastasis	7.2%	31.1%	<0.001
LRR per patient	12.5%	38.3%	<0.001
Time to local recurrence	13 months (IQR: 6–27)	19 months (IQR: 12–30)	0.64
Local control after salvage treatment	0/5	9/13	

Data are medians (IQR or CI). LRFS = local recurrence-free survival. PFS = progression-free survival. LRR = local recurrence rate.

## Data Availability

For reasons of patient privacy and consent, raw data are not available.

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
