# Peer review of "Colorectal Pulmonary Metastases: Pulmonary Metastasectomy or Stereotactic Radiotherapy?"

_cancers, 2023, doi:10.3390/cancers15215186_

Round 1
Reviewer 1 Report
van Drop et al. present a manuscript that compares resection of pulmonary CRC metastases to SABR. This is a retrospective study including to hospitals with different approaches to treatment of colorectal lung metastases. The topic in comparison is interesting as well as important, and there is lack of high-quality data comparing these the treatment methods.
Some comments about the paper
Methods:
The setting of having two major tertiary units with completely different treatment pathways makes the comparison possible, but raises questions: What are these differences based on? Are referring doctors and patients aware of these differences, and does this potentially lead to selection which would make the groups less comparable? This possible bias should be discussed in the manuscript. Do AUMC and MMC generally have equal results in oncology?
One of the main weaknesses of this study are the small group sizes. There is no propensity-matching (statistical matching) between the groups, and this is a clear weakness. I believe this is because of already quite small group sizes that propensity-matching probably would reduce further. The baseline characteristics are similar, but this does not really function as a proper substitute for matching. Did the authors experiment with a propensity score or some other form of matching with this material, and if yes, what were the results?
Also, as mentioned in the discussion, excluding resected patients with non-CRC pathology creates a bias, probably in favour of SABR. At least the specific diagnoses for the 6 patients excluded because of this should be clearly written in Figure 1.
Line 89-91: “Patients were excluded if they received local treatment for oligoprogression after previous palliative chemotherapy for polymetastatic conversion (n = 29)”. This clause is quite unclear, even though it is explained later in the manuscript. This oligoprogression happened mostly in SABR patients. Even if these patients were excluded, it raises the same concern that only “better” patients were referred to MMC for resection whereas also sicker patients were referred to AUMC for SABR.
What are the patients excluded for “no follow-up available”. Did they die before the first follow-up (should be included) or did they move abroad or something like that?
Background:
It would be more informative if right and left colonic primary tumours were reported separately (Table 1 & Appendix 1). More information on RAS and BRAF status would also be interesting, but that may not be available(?)
Outcome (methods and results):
OS is a difficult outcome when discussing CRC pulmonary metastases as most of the patients do not die from lung metastases, but from e.g., liver metastases or carcinomatosis. The PulMiCC study also showed long OS even without resection. One of the main benefits of operative/local treatments for pulmonary metastases is the possibility of curing the disesase. Disease recurrence or “PFS event” after local treatment may not be that important if the recurrence can be treated radically. This makes the chapter “3.4. Treatment of local failure” interesting. There was a clear difference in primary local control rates (failure in 12.5 vs. 38.3 %). It would be interesting to know whether there still would be a difference, when successful salvage therapies are were taken into account.
Discussion:
Generally, the Discussion part is reasonable, although some statements defending the current manuscript (as mentioned above) may not be so well motivated.
Conclusion:
I think this current study has a nice idea and story behind it, and it is well put together, but does not provide much evidence as such.
The main conclusion is that this whole study serves as base for forming hypothesis for future studies. This is clearly stated as it should be. It would be a highly interesting RCT!
This said, my opinion is that this manuscript is worth publishing after some revision.
The manuscript is well written and easy to read. "Patients were excluded if they received local treatment for oligoprogression after previous palliative chemotherapy for polymetastatic conversion (n = 29)" is really the only extremely annoying clause.
Author Response
Reviewer 1:
Comment 1:
van Dorp et al. present a manuscript that compares resection of pulmonary CRC metastases to SABR. This is a retrospective study including to hospitals with different approaches to treatment of colorectal lung metastases. The topic in comparison is interesting as well as important, and there is lack of high-quality data comparing these the treatment methods.
The setting of having two major tertiary units with completely different treatment pathways makes the comparison possible, but raises questions: What are these differences based on? Are referring doctors and patients aware of these differences, and does this potentially lead to selection which would make the groups less comparable? This possible bias should be discussed in the manuscript. Do AUMC and MMC generally have equal results in oncology?
One of the main weaknesses of this study are the small group sizes. There is no propensity-matching (statistical matching) between the groups, and this is a clear weakness. I believe this is because of already quite small group sizes that propensity-matching probably would reduce further. The baseline characteristics are similar, but this does not really function as a proper substitute for matching. Did the authors experiment with a propensity score or some other form of matching with this material, and if yes, what were the results?
Reply to comment 1:
We fully agree with the reviewer that selection bias comes into play with the current study design. Both centers are referral centers for the treatment of metastatic colorectal cancer, however, this does not prevent selection bias. Therefore, we have now included this in the limitation section (Line 338-339). We have tried to account for this by analysing many known prognostic factors including primary tumor, mutational status, patient characteristics, performance status, metastatic disease load, number and size of the metastases, and history of colorectal liver metastases (Table 1 and Appendix 1) for both groups and found no significant differences regarding these prognosticators. This was one of the major reasons why no statistical matching was performed, and in addition, as stated by the reviewer, this would further decrease the sample size (Discussion section Line 270-273).
Line 342-343: Inserted: We acknowledge the possibility of selection bias with this study design.
Comment 2:
Also, as mentioned in the discussion, excluding resected patients with non-CRC pathology creates a bias, probably in favor of SABR. At least the specific diagnoses for the 6 patients excluded because of this should be clearly written in Figure 1.
Reply to comment 2:
We agree this is an important point. Both Figure 1 and the corresponding text (Line 115-117), as well as the first paragraph of the discussion section (Line 273-277) and the limitations section of the discussion (Line 341-344) address this point. The final pathology of the individual patients is noted in Line 115-117. As suggested, we have now also added this information to Figure 1. Including these patients in the final analysis would not be prudent, as these patients did not have colorectal lung metastases. However, it is possible that similar patients ended up being included in the SABR group. In addition, not all local recurrences following SABR were pathologically proven. We have extensively reviewed all patient files, radiological images and multidisciplinary board review advice, and used this data in our manuscript. The authors have extensive experience in treating oligometastatic disease and analyzing its outcome, however, the frequent lack of pathology is a known downside of treatment with SABR (for smaller lesions) - this is inescapable.
Figure 1: Incorporated: Final pathology: no colorectal metastasis (e.g. non-small cell lung cancer, granuloma, focal atelectasis)
Comment 3:
Line 89-91: “Patients were excluded if they received local treatment for oligoprogression after previous palliative chemotherapy for polymetastatic conversion (n = 29)”. This clause is quite unclear, even though it is explained later in the manuscript. This oligoprogression happened mostly in SABR patients. Even if these patients were excluded, it raises the same concern that only “better” patients were referred to MMC for resection whereas also sicker patients were referred to AUMC for SABR.
Reply to comment 3:
During the study period patients with multiorgan metastatic colorectal cancer at the AUMC were also treated within the ORCHESTRA trial (NCT01792934) if they responded to palliative chemotherapy. These randomized patients were excluded from this report and it is mainly these patients that are in the group referred to by the reviewer.
Comment 4:
What are the patients excluded for “no follow-up available”. Did they die before the first follow-up (should be included) or did they move abroad or something like that?
Reply to comment 4:
We appreciate the opportunity to clarify this. These are not patients that died before the first follow-up. These patients would indeed not be excluded. They are patients who were mainly referred back to their referring physician in a different hospital, and if review of follow-up imaging was not possible then they were excluded from this analysis. We have clarified this in the revised article.
Line 116-117: Inserted: (e.g. no access to follow-up imaging elsewhere)
Comment 5:
It would be more informative if right and left colonic primary tumours were reported separately (Table 1 & Appendix 1). More information on RAS and BRAF status would also be interesting, but that may not be available(?)
Reply to comment 5:
We have added a line to the Table to separate right-sided to left-sided colon cancer. We also agree with the reviewer regarding mutation status. However, since we wanted to include patients with long-term follow-up, we included only patients who were treated from 2012 to 2019. During this timeframe, it was not standard of care to perform a thorough mutational status on all metastatic patients. Therefore, these outcomes are unfortunately not reported for most patients.
Table 1: Incorporated: Primary right-sided colon cancer 6 (15%) 13 (21.7%) 0.29
Primary left-sided colon cancer 15 (37.5%) 14 (23.3%)
Comment 6:
OS is a difficult outcome when discussing CRC pulmonary metastases as most of the patients do not die from lung metastases, but from e.g., liver metastases or carcinomatosis. The PulMiCC study also showed long OS even without resection. One of the main benefits of operative/local treatments for pulmonary metastases is the possibility of curing the disease. Disease recurrence or “PFS event” after local treatment may not be that important if the recurrence can be treated radically. This makes the chapter “3.4. Treatment of local failure” interesting. There was a clear difference in primary local control rates (failure in 12.5 vs. 38.3 %). It would be interesting to know whether there still would be a difference, when successful salvage therapies are were taken into account.
Reply to comment 6:
We fully agree that the outcome of local failure is very interesting. Therefore, we also incorporated Figure 5 to demonstrate the value of local salvage therapy and analyzed the outcomes of the different salvage treatments. To answer the question of the reviewer, 9 of the 23 patients who developed a local recurrence following SABR also developed other distant progression.
Comment 7:
Generally, the Discussion part is reasonable, although some statements defending the current manuscript (as mentioned above) may not be so well motivated. I think this current study has a nice idea and story behind it, and it is well put together, but does not provide much evidence as such. The main conclusion is that this whole study serves as base for forming hypothesis for future studies. This is clearly stated as it should be. It would be a highly interesting RCT! This said, my opinion is that this manuscript is worth publishing after some revision.
Reply to comment 7:
We thank the author for the kind remarks, and we will use the comments and experience gained from this report to further optimize the design of the proposed RCT.
Comment 8:
The manuscript is well written and easy to read. "Patients were excluded if they received local treatment for oligoprogression after previous palliative chemotherapy for polymetastatic conversion (n = 29)" is really the only extremely annoying clause.
Reply to comment 8:
Thank you – we have adjusted this and hope it is better.
Line 109-110: Changed to: Patients were excluded if they were treated for oligoprogression after previous palliative chemotherapy for polymetastatic conversion (n = 29).
Reviewer 2 Report
I recommend authors to focus in presenting the novelty and the importance of this study
The study period is up to 2019 why do not you include patients after 2019 as the study is retrospective?
In methods kindly add reference for the follow-up period you consider
The new lesions you recorded, I recommend providing the duration of each mets appeared
Fine
Author Response
Reviewer 2:
Comment 1:
I recommend authors to focus in presenting the novelty and the importance of this study. The study period is up to 2019 why do not you include patients after 2019 as the study is retrospective? In methods kindly add reference for the follow-up period you consider.
Reply to comment 1:
Thank you. For this analysis we really wanted to have long term follow-up for all patients, therefore we included only patients that were treated from 2012 to 2019.
Comment 2:
The new lesions you recorded, I recommend providing the duration of each mets appeared.
Reply to comment 2:
Many thanks. We have reported the median time to local recurrence for both groups in the results section, Line 219-220 and Line 230. The median time to disease progression for both groups are included in Line 202-204.
Comment 3:
Comments on the Quality of English Language: Fine
Reply to comment 3:
We thank the author for the kind remarks.
Reviewer 3 Report
This manuscript is an original article that retrospectively compared pulmonary metastasectomy and stereotactic ablative radiotherapy (SABR) in patients with colorectal pulmonary metastases (CRPM) eligible for local treatment.
The authors showed that pulmonary metastasectomy and SABR had comparable overall survival, local recurrence-free survival, and complication rates, despite patients in the SABR group having a significantly lower progression-free survival and local control rate.
This study was conducted well, and the data are presented clearly. I think this is a well-written paper with interesting data.
The results will be of interest to clinicians and researchers in the field.
The following a minor issue require clarification:
Minor
1. (Abstract) Please provide unabbreviated words of “LRFS” and “PFS”.
2. I recommend that that the authors describe the irradiation condition and planning in detail in the Method section.
Author Response
Reviewer 3:
Comment 1:
This manuscript is an original article that retrospectively compared pulmonary metastasectomy and stereotactic ablative radiotherapy (SABR) in patients with colorectal pulmonary metastases (CRPM) eligible for local treatment. The authors showed that pulmonary metastasectomy and SABR had comparable overall survival, local recurrence-free survival, and complication rates, despite patients in the SABR group having a significantly lower progression-free survival and local control rate.
This study was conducted well, and the data are presented clearly. I think this is a well-written paper with interesting data. The results will be of interest to clinicians and researchers in the field.
Reply to comment 1:
We thank the author for the kind and positive remarks.
Comment 2:
(Abstract) Please provide unabbreviated words of “LRFS” and “PFS”.
Reply to comment 2:
Line 48: Changed to: local recurrence-free survival (LRFS)
Line 49: Changed to: progression-free survival (PFS)
Comment 3:
I recommend that that the authors describe the irradiation condition and planning in detail in the Method section.
Reply to comment 3:
See the reply to the comment from the editor.
Reviewer 4 Report
It is very honored to be invited the reviewing the manuscript “Colorectal pulmonary metastases: pulmonary metastasectomy or stereotactic radiotherapy?”
This manuscript is well written and the substance and conclusion in this paper were interesting and clinically meaningful.
The authors described the aggressive use of local salvage therapy might influenced on favorable OS in the patients with local recurrence after SABR. However, it is also a fact that the recent systemic chemotherapies improved survival especially in patients with colorectal cancer. Therefore, the authors should stated the above comment in the discussion.
Author Response
Reviewer 4:
Comment 1:
It is very honored to be invited the reviewing the manuscript “Colorectal pulmonary metastases: pulmonary metastasectomy or stereotactic radiotherapy?”. This manuscript is well written and the substance and conclusion in this paper were interesting and clinically meaningful.
The authors described the aggressive use of local salvage therapy might influenced on favorable OS in the patients with local recurrence after SABR. However, it is also a fact that the recent systemic chemotherapies improved survival especially in patients with colorectal cancer. Therefore, the authors should stated the above comment in the discussion.
Reply to comment 1:
We thank the author for the kind remarks.
We fully agree with the comment of the reviewer. We have incorporated this message into the limitations section of the discussion.
Line 336-337: Inserted: Improvements in systemic therapy over time may influence parameters like OS in future cohorts.
Round 2
Reviewer 1 Report
Thank you for the authors for answering all my questions. I think the quality of the manuscript has improved, and I would happy to support accepting it.